# Stable Hadamard Memory: Revitalizing Memory-Augmented Agents for Reinforcement Learning

**Hung Le, Dung Nguyen, Kien Do, Sunil Gupta, and Svetha Venkatesh**
Applied AI Institute, Deakin University, Geelong, Australia
`{thai.le,dung.nguyen,k.do,sunil.gupta,svetha.venkatesh}@deakin.edu.au`

## Abstract

Effective decision-making in partially observable environments demands robust memory management. Despite their success in supervised learning, current deep-learning memory models struggle in reinforcement learning environments that are partially observable and long-term. They fail to efficiently capture relevant past information, adapt flexibly to changing observations, and maintain stable updates over long episodes. We theoretically analyze the limitations of existing memory models within a unified framework and introduce the Stable Hadamard Memory, a novel memory model for reinforcement learning agents. Our model dynamically adjusts memory by erasing no longer needed experiences and reinforcing crucial ones computationally efficiently. To this end, we leverage the Hadamard product for calibrating and updating memory, specifically designed to enhance memory capacity while mitigating numerical and learning challenges. Our approach significantly outperforms state-of-the-art memory-based methods on challenging partially observable benchmarks, such as meta-reinforcement learning, long-horizon credit assignment, and POPGym, demonstrating superior performance in handling long-term and evolving contexts. Our source code is available at `https://github.com/thaihungle/SHM`.

## 1 Introduction

Reinforcement learning agents necessitate memory. This is especially true in Partially Observable Markov Decision Processes (POMDPs (Kaelbling et al., 1998)), where past information is crucial for making informed decisions. However, designing a robust memory remains an enduring challenge, as agents must not only store long-term memories but also dynamically update them in response to evolving environments. Memory-augmented neural networks (MANNs)–particularly those developed for supervised learning (Graves et al., 2016; Vaswani et al., 2017), while offering promising solutions, have consistently struggled in these dynamic settings. Recent empirical studies (Morad et al., 2023; Ni et al., 2024) have shown that MANNs exhibit instability and underperform simpler vector-based memory models such as GRU (Chung et al., 2014) or LSTM (Hochreiter, 1997). The issue is exacerbated in complex and sparse reward scenarios where agents must selectively retain and erase memories based on relevance. Unfortunately, existing methods fail to provide a memory writing mechanism that is efficient, stable, and flexible to meet these demands.

In this paper, we focus on designing a better writing mechanism to encode new information into the memory. To this end, we introduce the Hadamard Memory Framework (HMF), a unified model that encompasses many existing writing methods as specific cases. This framework highlights the critical role of *memory calibration*, which involves linearly adjusting memory elements by multiplying the memory matrix with a *calibration matrix* and then adding an *update matrix*. By leveraging Hadamard products that operate element-wise on memory matrices, we allow memory writing without mixing the memory cells in a computationally efficient manner. More importantly, the calibration and update matrices are dynamically computed based on the input at each step. This enables the model to learn adaptive memory rules, which are crucial for generalization. For instance, in meta-reinforcement learning with varying maze layouts, a fixed memory update rule may work for one layout but fail in another. By allowing the calibration matrix to adjust according to the current

maze observation, the agent can learn to adapt to any layout configuration. A dynamic calibration matrix also enables the agent to flexibly forget and later recall information as needed. For example, an agent navigating a room may need to remember the key's location, retain it during a detour, and later recall it when reaching a door, while discarding irrelevant detour events.

Although most current memory models can be reformulated within the HMF, they tend to be either overly simplistic with limited calibration capabilities (Katharopoulos et al., 2020; Radford et al., 2019) or unable to manage memory writing reliably, suffering from gradient vanishing or exploding issues (Ba et al., 2016; Morad et al., 2024). To address these limitations, we propose a specific instance of HMF, called Stable Hadamard Memory, which introduces a novel calibration mechanism based on two key principles: (i) dynamically adjusting memory values in response to the current context input to selectively weaken outdated or enhance relevant information, and (ii) ensuring the expected value of the calibration matrix product remains bounded, thereby preventing gradient vanishing or exploding. Through extensive experimentation on POMDP benchmarks, including meta reinforcement learning, long-horizon credit assignment, and hard memorization games, we demonstrate that our method consistently outperforms state-of-the-art memory-based models in terms of performance while also delivering competitive speed. We also provide comprehensive ablation studies that offer insights into the components and internal workings of our memory models.

In summary, our contributions are: **(i) Unified Memory Framework:** We propose a unified memory framework, supported by theoretical analysis, to elucidate its properties as episode length increases. **(ii) Novel Calibration Matrix:** We design a calibration matrix with random parameter selection, mitigating timestep dependencies and theoretically enhancing stability by regulating the expected gradient norm. **(iii) Extensive Empirical Evaluation:** Experiments on challenging benchmarks demonstrate the scalability and superior performance of our approach across various tasks.

## 2 BACKGROUND

### 2.1 REINFORCEMENT LEARNING PRELIMINARIES

A Partially Observable Markov Decision Process (POMDP) is formally defined as a tuple $\langle S, A, O, R, P, \gamma \rangle$, where $S$ is the set of states, $A$ is the set of actions, $O$ is the observation space, $R : S \times A \to \mathbb{R}$ is the reward function, $P : S \times A \to \Delta(S)$ defines the state transition probabilities, and $\gamma \in [0, 1)$ is the discount factor. Here, the agent does not directly observe the true environment state $s_t$. Instead, it receives an observation $o_t \sim O(s_t)$, which provides partial information about the state, often not enough for optimal decision making. Therefore, the agent must make decisions based on its current observation $o_t$ and a history of previous observations, actions, and rewards $(a_0, r_0, o_1, \ldots, a_{t-1}, r_{t-1}, o_t)$. The history may exclude past rewards or actions.

Let us denote the input context at timestep $t$ as $x_t = (o_t, a_{t-1}, r_{t-1})$ and assume that we can encode the sequence of contexts into a memory $M_t = f\left(\{x_i\}_{i=1}^t\right)$. The goal is to learn a policy $\pi(a_t|M_t)$ that maximizes the expected cumulative discounted reward:

$$J(\pi) = \mathbb{E}_\pi \left[ \sum_{t=1}^\infty \gamma^t R(s_t, a_t) | a_t \sim \pi(a_t|M_t), s_{t+1} \sim P(s_{t+1}|s_t, a_t), o_t \sim O(s_t) \right] \quad (1)$$

Thus, a memory system capable of capturing past experiences is essential for agents to handle the partial observability of the environment while maximizing long-term rewards.

### 2.2 MEMORY-AUGMENTED NEURAL NETWORKS

We focus on matrix memory $M$, and to simplify notation, we assume it is a square matrix. Given a memory $M \in \mathbb{R}^{H \times H}$, we usually read from the memory as:

$$h_t = M_t q\left(x_t\right) \quad (2)$$

where $q$ is a query network $q : \mathbb{R}^D \mapsto \mathbb{R}^H$ to map an input context $x_t \in \mathbb{R}^D$ to a query vector. The read value $h_t$ later will be used as the input for policy/value functions. Even more important than the reading process is the memory writing: *How can information be written into the memory*

*to ensure efficient and accurate memory reading?* A general formulation for memory writing is $M_t = f(M_{t-1}, x_t)$ with $f$ as the update function that characterizes the memory models.

The simplest form of memory writing traces back to Hebbian learning rules: $M_t = M_{t-1} + x_t \otimes x_t$ where $\otimes$ is outer product (Kohonen & Ruohonen, 1973; Hebb, 2005). Later, researchers have proposed "fast weight" memory (Marr & Thach, 1991; Schmidhuber, 1992; Ba et al., 2016):

$$M_t = M_{t-1}\lambda + \eta g(M_{t-1}, x_t) \otimes g(M_{t-1}, x_t) \tag{3}$$

where $g$ is a non-linear function that take the previous memory and the current input data as the input; $\lambda$ and $\eta$ are constant hyperparameters. On the other hand, computer-inspired memory architectures such as Neural Turing Machine (NTM, (Graves et al., 2014)) and Differentiable Neural Computer (DNC, (Graves et al., 2016)) introduce more sophisticated memory writing:

$$M_t = M_{t-1} \odot (\mathbf{1} - w(M_{t-1}, x_t) \otimes e(M_{t-1}, x_t)) + w(M_{t-1}, x_t) \otimes v(M_{t-1}, x_t) \tag{4}$$

where $w$, $e$ and $v$ are non-linear functions that take the previous memory and the current input data as the input to produce the writing weight, erase and value vectors, respectively. $\odot$ is the Hadamard (element-wise) product. The problem with non-linear $f$ w.r.t $M$ is that the computation must be done in recursive way, and thus being slow. Therefore, recent memory-based such as Linear Transformer models adopt simplified linear update (Katharopoulos et al., 2020):

$$M_t = M_{t-1} + v(x_t) \otimes \frac{\phi(k(x_t))}{\sum_t \phi(k(x_t))} \tag{5}$$

where $\phi$ is an activation function; $k$ and $v$ are functions that transform the input to key and value. In another perspective inspired by neuroscience, Fast Forgetful Memory (FFM, (Morad et al., 2024)) employs a parallelable memory writing, which processes a single step update as follows:

$$M_t = M_{t-1} \odot \gamma + \left(v(x_t) \otimes \mathbf{1}^\top\right) \odot \gamma^{t-n} \tag{6}$$

where $\gamma$ is a trainable matrix, $v$ is an input transformation function, and $n$ is the last timestep (see more memory models in Appendix D).

## 3 METHODS

In this section, we begin by introducing a unified memory writing framework that incorporates several of the memory writing approaches discussed earlier. Next, we examine the limitations of current memory writing approaches through an analysis of this framework. Following this analysis, we propose specialized techniques to address these limitations. For clarity and consistency, all matrix and vector indices will be referenced starting from 1, rather than 0. Constant matrices are denoted by bold numbers.

### 3.1 HADAMARD MEMORY FRAMEWORK (HMF)

We propose a general memory framework that uses the Hadamard product as its core operation. The memory writing at time step $t$ is defined as:

$$M_t = M_{t-1} \odot \underbrace{C_\theta(x_t)}_{C_t} + \underbrace{U_\varphi(x_t)}_{U_t} \tag{7}$$

where $C_\theta : \mathbb{R}^D \mapsto \mathbb{R}^{H \times H}$ and $U_\varphi : \mathbb{R}^D \mapsto \mathbb{R}^{H \times H}$ are parameterized functions that map the current input $x_t$ to two matrices $C_t$ (calibration matrix) and $U_t$ (update matrix). Here, $\theta$ and $\varphi$ are referred to as calibration and update parameters. Intuitively, the calibration matrix $C_t$ determines which parts of the previous memory $M_{t-1}$ should be weakened and which should be strengthened while the update matrix $U_t$ specifies the content to be encoded into the memory. We specifically choose the Hadamard product ($\odot$) as the matrix operator because it operates on each memory element individually. We avoid using the matrix product to prevent mixing the content of different memory cells during calibration and update. Additionally, the matrix product is computationally slower.

There are many ways to design $C_t$ and $U_t$. Given proper choices of $C_t$ and $U_t$, Eqs. 3-6 can be reformulated into Eq. 7. Inspired by prior "fast weight" works, we propose a simple update matrix:

$$U_\varphi(x_t) = \eta_\varphi(x_t)[v(x_t) \otimes k(x_t)]$$

(8)

where $k$ and $v$ are trainable neural networks that transform the input $x_t$ to key and value representations. $\eta_\varphi : \mathbb{R}^D \mapsto \mathbb{R}$ is a parameterized function that maps the current input $x_t$ to an update gate that controls the amount of update at step $t$. For example, if $\eta_\varphi(x_t) = 0$, the memory will not be updated with any new content, whereas $\eta_\varphi(x_t) = 1$ the memory will be fully updated with the content from the $t$-th input like Linear Transformer. We implement $\eta_\varphi(x_t)$ as a neural network with sigmoid activation function.

We now direct our focus towards the design of $C_t$, which is the core contribution of our work. The calibration matrix selectively updates the memory by erasing no longer important memories and reinforcing ongoing critical ones. In a degenerate case, if $C_t = \mathbf{1}$ for all $t$, the memory will not forget or strengthen any past information, and will only memorize new information over time, similar to the Hebbian Rule and Linear Transformer. To analyze the role of the calibration matrix, it is useful to unroll the recurrence, leading to the closed-form equation (see proof in Appendix A.1):

$$M_t = M_0 \prod_{i=1}^{t} C_i + \sum_{i=1}^{t} U_i \odot \prod_{j=i+1}^{t} C_j$$

(9)

where $\prod$ represents element-wise products. Then, $h_t = M_t q(x_t) = M_0 \prod_{i=1}^{t} C_i q(x_t) + \sum_{i=1}^{t} U_i \odot \prod_{j=i+1}^{t} C_j q(x_t)$. Calibrating the memory is important because without calibration ($C_t = \mathbf{1} \,\forall t$), the read value becomes: $h_t = M_0 q(x_t) + \sum_{i=1}^{t} U_i q(x_t)$. In this case, making $h_t$ to reflect a past context at any step $j$ requires that $q(x_t) \neq 0$ and $\sum_{i \neq j} U_i q(x_t) = \sum_{i \neq j} \eta_\varphi(x_i) v(x_i)[k(x_i) \cdot q(x_t)] \approx \mathbf{0}$, which can be achieved if we can find $q(x_t)$ such that $k(x_i) \cdot q(x_t) \approx \mathbf{0} \,\forall i \neq j$. Yet, this becomes hard when $T \gg H$ and $\eta_\varphi(x_i) \neq \mathbf{0}$ as it leads to an overdetermined system with more equations than variables. We note that avoiding memorizing any $i$-th step with $\eta_\varphi(x_i) = 0$ is suboptimal since $x_i$ may be required for another reading step $t' \neq t$.

Therefore, at a certain timestep $t$, it is critical to eliminate no longer relevant timesteps from $M_t$ by calibration, i.e., $U_i \odot \prod_{j=i+1}^{t} C_j \approx \mathbf{0}$ for unimportant $i$ (forgetting). For example, an agent may first encounter an important event, like seeing a color code, before doing less important tasks, such as picking apples. When reaching the goal requiring to identify a door matching the color code, it would be beneficial for the agent to erase memories related to the apple-picking task, ensuring a clean retrieval of relevant information–the color code. Conversely, if timestep $i$ becomes relevant again at a later timestep $t'$, we need to recover its information, ensuring $U_i \odot \prod_{j=i+1}^{t'} C_j \neq \mathbf{0}$ (strengthening), just like the agent, after identifying the door, may return to collecting apple task.

*Remark* 1. In the Hadamard Memory Framework, calibration should be enabled ($C_t \neq \mathbf{1}$) and conditioned on the input context.

Regarding computing efficiency, if $C_t$ and $U_t$ are not functions of $M_{<t}$, we can compute the memory using Eq. 9 in parallel, ensuring fast execution. In particular, the set of products $\left\{ \prod_{j=i+1}^{t} C_j \right\}_{i=1}^{t}$ can be calculated in parallel in $O(\log t)$ time. The summation can also be done in parallel in $O(\log t)$ time. Additionally, since all operations are element-wise, they can be executed in parallel with respect to the memory dimensions. Consequently, the total time complexity is $O(\log t)$. Appendix Algo. 1 illustrates an implementation supporting parallelization.

*Remark* 2. In the Hadamard Memory Framework, with optimal parallel implementation, the time complexity for processing a sequence of $t$ steps is $O(\log t)$. By contrast, without parallelization, the time complexity is $O(tH^2)$.

### 3.2 CHALLENGES ON MEMORY CALIBRATION

The calibration matrix enables agents to either forget or enhance past memories. However, it complicates learning due to the well-known issues of gradient vanishing or exploding. This can be observed when examining the policy gradient over $T$ steps, which reads:

$$\nabla_\Theta J\left(\pi_\Theta\right) = \mathbb{E}_{s,a\sim\pi_\Theta} \sum_{t=0}^{T} \underbrace{\nabla_\Theta \log \pi_\Theta\left(a_t|M_t\right)}_{G_t(\Theta)} Adv\left(s_t, a_t, \gamma\right) \tag{10}$$

where $\Theta$ is the set of parameters, containing $\{\theta, \varphi\}$, $Adv$ represents the advantage function, which integrates reward information $R\left(s_t, a_t\right)$, and $G_t\left(\Theta\right) = \frac{\partial \log \pi_\Theta\left(a_t|M_t\right)}{\partial M_t} \frac{\partial M_t}{\partial \Theta}$ captures information related to the memory. Considering main gradient at step $t$, required to learn $\theta$ and $\varphi$:

$$\frac{\partial M_t}{\partial \theta} = M_0 \frac{\partial \prod_{i=1}^{t} C_\theta(x_i)}{\partial \theta} + \sum_{i=1}^{t} U_\varphi\left(x_i\right) \odot \underbrace{\frac{\partial \prod_{j=i+1}^{t} C_\theta(x_j)}{\partial \theta}}_{G_1(i,t,\theta)}; \tag{11}$$

$$\frac{\partial M_t}{\partial \varphi} = \sum_{i=1}^{t} \frac{\partial U_\varphi\left(x_i\right)}{\partial \varphi} \odot \underbrace{\prod_{j=i+1}^{t} C_\theta(x_j)}_{G_2(i,t,\theta)} \tag{12}$$

We collectively refer $G_1\left(i, t, \theta\right)$ and $G_2\left(i, t, \theta\right)$ as $G_{1,2}\left(i, t, \theta\right) \in \mathbb{R}^{H \times H}$. These terms are critical as they capture the influence of state information at timestep $i$ on the learning parameters $\theta$ and $\varphi$. The training challenges arise from these two terms as the number of timesteps $t$ increases: (i) *Numerical Instability (Gradient Exploding):* if $\exists m, k \in [1, H]$ s.t. $G_{1,2}\left(i, t, \theta\right)[m, k] \to \infty$, this leads to overflow, causing the gradient to become "nan"; (ii) *Learning Difficulty (Gradient Vanishing)*: if $t \gg i_0$, $\|G_{1,2}\left(i, t, \theta\right)\| \approx 0 \ \forall i < i_0$, meaning no signal from timesteps $i < i_0$ contributes to learning the parameters. This is suboptimal, especially when important observations occur early in the episode, and rewards are sparse and given at the episode end, .i.e., $R\left(s_t, a_t\right) = 0 \ \forall t \neq T$.

*How to design the calibration matrix $C_\theta\left(x\right)$ to overcome the training challenges?* A common approach is to either fix it as hyperparameters or make it learnable parameters independent on the input $x_t$ (e.g., Eqs. 3 and 6). Unfortunately, we can demonstrate that this leads to either numerical instability or learning difficulties as formalized in Proposition 3. In the next section, we will provide a better design for the calibration matrix.

**Proposition 3.** *If calibration is enabled such that $C_\theta(x_t) \neq 1$, and the calibration matrix remains fixed and independent of the input $x_t$ (i.e., $\forall t : C_\theta(x_t) = \theta \in \mathbb{R}^{H \times H}$), this will lead to either numerical instability or learning difficulties.*

*Proof.* See Appendix A.2 □

### 3.3 STABLE HADAMARD MEMORY (SHM)

To avoid numerical and learning problems, it is important to ensure each element of $C_t$ is not always greater than 1 or smaller than 1, which ends up in their product will be bounded such that $\mathbb{E}\left[\prod_{t=1}^{T} C_t\right] \neq \{\mathbf{0}, \infty\}$ as $T \to \infty$. At the same time, we want $C_t$ to be a function of $x_t$ to enable calibration conditioned on the current context. To this end, we propose the calibration matrix:

$$\boxed{C_\theta\left(x_t\right) = 1 + \tanh\left(\theta_t \otimes v_c\left(x_t\right)\right)} \tag{13}$$

where $v_c : \mathbb{R}^D \mapsto \mathbb{R}^H$ is a mapping function, and $\theta_t \in \mathbb{R}^H$ represents the main calibration parameters. Here, we implement $v_c$ as a linear transformation to map the input to memory space.

Notably, the choice of $\theta_t$ determines the stability of the calibration. *We propose to design $\theta_t$ as trainable parameters that is randomly selected from a set of parameters $\theta$.* In particular, given $\theta = \{\theta_l\}_{l=1}^{L}$, where $\theta_l \in \mathbb{R}^H$ and $L$ is the set size, we sample uniformly a random parameter from $\theta$: $\theta_t \sim \mathcal{U}(\theta)$. We name the design as Stable Hadamard Memory (SHM). Given the formulation, the range of an element $z_t^{m,k} = C_\theta(x_t)[m,k]$ is $[0,2]$ where $m,k \in [1,H]$. We can show one important property of this choice is that $\mathbb{E}\left[\prod_{t=1}^{T} C_t\right] \approx \mathbf{1}$ under certain conditions.

**Proposition 4.** *Assume: (i) $x_t \sim \mathcal{N}(0,\Sigma_t)$, (ii) $v_f$ is a linear transformation, i.e.,$v_c(x_t) = Wx_t$, (iii) $\{z_t = \theta_t \otimes v_c(x_t)\}_{t=1}^{T}$ are independent across $t$. Given $C_\theta(x_t)$ defined in Eq. 13 then :*

$$\mathbb{E}\left[\prod_{t=1}^{T} z_t^{m,k}\right] = \mathbb{E}\left[\prod_{t=1}^{T} C_\theta(x_t)[m,k]\right] = 1 \ \forall T \geq 0, 1 \leq m,k \leq H$$

*Proof.* see Appendix A.3 □

Assumption (i) is reasonable as $x_t$ can be treated as being drawn from a Gaussian distribution, and LayerNorm can be applied to normalize $x_t$ to have a mean of zero. Assumption (ii) can be realized as we implement $v_c$ as a linear transformation. Assumption (iii) is more restrictive because $\{x_t\}_{t=1}^{T}$ are often dependent in RL setting, which means $\{z_t = \theta_t \otimes v_c(x_t)\}_{t=1}^{T}$ are not independent and thus, $\mathbb{E}\left[\prod_{t=1}^{T} z_t^{m,k}\right] \neq 1$. However, by reducing the linear dependence between $z_t$ through random selection of $\theta_t$, we can make $\mathbb{E}\left[\prod_{t=1}^{T} z_t^{m,k}\right]$ closer to 1, and thus being bounded. Specifically, we will prove that by using $\theta_t \sim \mathcal{U}(\theta)$, the Pearson Correlation Coefficient between timesteps is minimized, as stated in the following proposition:

**Proposition 5.** *Let $z_t^{m,k} = u_t^m v_t^k$ where $z_t^{m,k} = (\theta_t \otimes v_c(x_t))[m,k]$, $u_t^m = \theta_t[m]$ and $v_t^k = v_c(x_t)[k]$. Given the Pearson correlation coefficient of two random variables $X$ and $Y$ is defined as $\rho(X,Y) = \frac{Cov(X,Y)}{\sqrt{Var(X)}\sqrt{Var(Y)}}$, then $\forall v_t^k, v_{t'}^k$:*

$$\left|\rho\left(u_t^m v_t^k, u_{t'}^m v_{t'}^k\right)\right| \leq \left|\rho\left(v_t^k, v_{t'}^k\right)\right|$$

*Proof.* See Appendix A.4 □

As a result, our choice of $\theta_t$ outperforms other straightforward designs for minimizing dependencies between timesteps. For instance, a fixed $\theta_t$ ($\theta_t = \theta \in \mathbb{R}^H$) results in higher dependencies because $\rho(\theta[m]v_t^k, \theta[m]v_{t'}^k) = \rho(v_t^k, v_{t'}^k) \geq \rho\left(u_t^m v_t^k, u_{t'}^m v_{t'}^k\right)$. In practice, even when $\mathbb{E}\left[\prod_{t=1}^{T} z_t^{m,k}\right]$ is bounded, the cumulative product can occasionally become very large for certain episodes and timesteps, leading to overflow and disrupting the learning process. This can be avoided by clipping the gradients. In experiments, we implement SHM using nonparallel recursive form (Eq. 7). The memory is then integrated into policy-gradient RL algorithms to optimize Eq. 1, with the read value $h_t$ (Eq. 2) used as input for value/policy networks.

The proposed update rule improves memory management and stability through several key benefits. The dynamic calibration matrix $C_t$ adjusts memory elements based on the input context, allowing the model to prioritize relevant data while forgetting outdated information. This ensures that transient details, like detour events, are discarded while critical information, like landmarks, is retained. Moreover, the random parameter selection in the calibration matrix breaks timestep dependencies and controls the expected gradient norm, preventing numerical instabilities like gradient explosion or vanishing, and ensureing robust performance over long episodes.

## 4 EXPERIMENTAL RESULTS

We evaluate our method alongside notable memory-augmented agents in POMDP environments. Unless stated otherwise, the context consists of the observation and previous action. All training

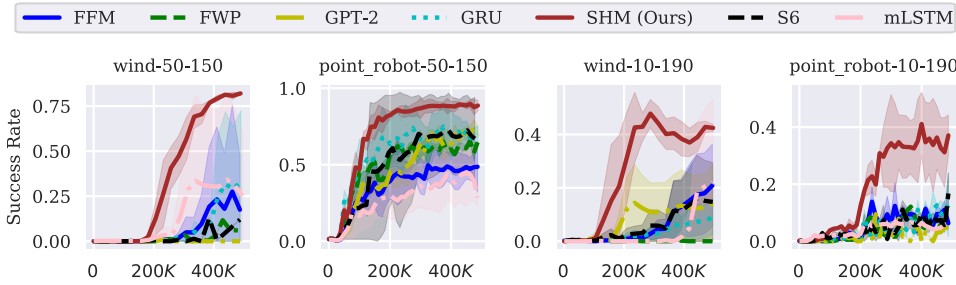

Figure 1: Meta-RL: Wind and Point Robot learning curves. Mean ± std. over 5 runs.

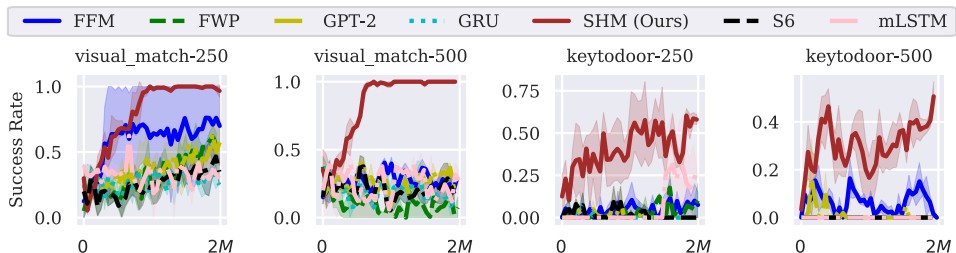

Figure 2: Credit Assignment: Visual Match, Key-to-Door learning curves. Mean ± std. over 3 runs.

uses the same hardware (single NVDIA H100 GPU), RL architecture, algorithm, training protocol, and hyperparameters. The baselines differ only in their memory components: GRU (Chung et al., 2014), FWP (Schlag et al., 2021), GPT-2 (Radford et al., 2019), S6 (Gu & Dao, 2023), mLSTM (Beck et al., 2024b), FFM (Morad et al., 2024) and SHM (Ours). For tasks with a clear goal, we measure performance using the Success Rate, defined as the ratio of episodes that reach the goal to the total number of evaluation episodes. For tasks in POP-Gym, we use Episode Return as the evaluation metric. We fix SHM's number of possible $\theta_t$, $L = 128$, across experiments.

## 4.1 SAMPLE-EFFICIENT META REINFORCEMENT LEARNING

Meta-RL targets POMDPs where rewards and environmental dynamics differ across episodes, representing various tasks (Schmidhuber, 1987; Thrun & Pratt, 1998). To excel in all tasks, memory agents must learn general memory update rules that can adapt to any environments. We enhanced the Wind and Point Robot environments from Ni et al. (2022) to increase difficulty. In these environments, the observation consists of the agent's 2D position $p_t$, while the goal state $p_g$ is hidden. The agent takes continuous actions $a_t$ by moving with 2D velocity vector. The sparse reward is defined as $R(p_{t+1}, p_g) = \mathbf{1}(\|p_{t+1} - p_g\|_2 \leq r)$ where $r = 0.01$ is the radius. In Wind, the goal is fix $p_g = [0, 1]$, yet there are noises in the dynamics: $p_{t+1} = p_t + a_t + w$, with the "wind" $w$ sampled from $U[-0.1, 0.1]$ at the start and fixed thereafter. In Point Robot, the goal varies across episodes, sampled from $U[-10, 10]$. To simulate real-world conditions where the training tasks are limited, we create 2 modes using different number of training and testing tasks: $[50, 150]$ and $[10, 190]$, respectively. Following the modifications, these simple environments become significantly more challenging to navigate toward the goal, so we set the episode length to 100.

We incorporate the memory methods to the Soft Actor Critic (SAC, (Haarnoja et al., 2018)), using the same code base introduced in Ni et al. (2022). We keep the SHM model sizes and memory capacities small, at around 2MB for the checkpoint and 512 memory elements, which is roughly equivalent to a GRU (see Appendix C.1). We train all models for 500,000 environment steps, and report the learning curves in Fig. 1. In the easy mode (50-150), our SHM method consistently achieves the best performance, with a near-optimal success rate, while other methods underperform by 20-50% on average in Wind and Point Robot, respectively. In the hard mode (10-190), SHM continues to outperform other methods by approximately 20%, showing earlier signs of learning.

| Model | GRU | FFM | SHM (Ours) |
|---|---|---|---|
| Average Return | -28.4±1.3 | -24.2±1.2 | **-5.1±6.3** |

Table 1: PopGym Hardest Tasks: Mean return ± std. ($\times 100$) at the end of training over 3 runs. The range of return ($\times 100$) is $[-100, 100]$.

## 4.2 LONG-TERM CREDIT ASSIGNMENT

In this task, we select the Visual Match and Key-to-Door environments, the most challenge tasks mentioned in Ni et al. (2024). Both have observation as the local view of the agent, discrete actions and sparse rewards dependent on the full trajectory, requiring long-term memorization. In particular, the pixel-based Visual Match task features an intricate reward system: in Phase 1, observing color codes yields no rewards, while in Phase 2, picking an apple provides immediate reward of one, relying on short-term memory. The final reward–a bonus for reaching the door with the matching color code is set to 10. Key-to-Door also involves multiple phases: finding the key, picking apples, and reaching the door. The terminal reward is given if the key was acquired in the first phase and used to open the door. Both tasks can be seen as decomposed episodic problems with noisy immediate rewards, requiring that in the final phase, the agent remembers the event in the first phase. We create 2 configurations using different episode steps in the second phases: 250 and 500, respectively.

We use SAC-Discrete (Christodoulou, 2019) as the RL algorithm. We train the same set of baselines as in Sec. 4.1 for 2 million environment steps. The results in Fig. 2 show that SHM significantly outperforms all other methods in success rate. Notably, SHM is the only method that can perfectly solve both 250 and 500-step Visual Match while the second-best performer, FFM, achieves only a 77% and 25% success rate, respectively. In Key-To-Door, our method continues showing superior results with high success rate. By contrast, no meaningful learning is observed from the others, which perform similarly to GPT-2, as also noted by Ni et al. (2024).

## 4.3 POPGYM HARDEST GAMES

We evaluate SHM on the POPGym benchmark (Morad et al., 2023), the largest POMDP benchmark to date. Following previous studies (Morad et al., 2023; Samsami et al., 2024), we focus on the most memory-intensive tasks: Autoencode, Battleship, Concentration and RepeatPrevious. These tasks require ultra long-term memorization, with complexity increasing across Easy, Medium, and Hard levels. All tasks use categorical action and observation spaces, allowing up to 1024 steps.

For comparison, we evaluate SHM against *state-of-the-art model-free methods*, including GRU and FFM. Other memory models, such as DNC, Transformers, FWP, and SSMs, have been reported to perform worse. The examined memory models are integrated into PPO (Schulman et al., 2017), trained for 15 million steps using the same codebase as Morad et al. (2023) to ensure fairness. The models differ only in their memory, controlled by the memory dimension hyperparameter $H$. We tune it for each baseline, adjusting it to optimize performance, as larger $H$ values typically improve results. The best-performing configurations are reported in Table 1 and Appendix Table 6, where SHM demonstrates a relative improvement of $\approx$10-12% over FFM and GRU on average. Notably, the learning curves in Appendix Fig. 4 show that only SHM demonstrates signs of learning in several tasks, including RepeatPrevious-Medium/Hard, and Autoencode-Easy/Medium. Detailed hyperparameter setting and additional results are provided in Appendix C.2.

In terms of running time, the average batch inference time in milliseconds for GRU, FFM, and SHM is 1.6, 1.8, and 1.9, respectively, leading to a total of 7, 8, and 9 hours of training per task. While SHM is slightly slower than GRU and FFM, the difference is a reasonable trade-off for improved memory management in partially observable environments. Last but not least, our model's runtime was measured using a non-parallel implementation, while GRU benefits from hardware optimization in the PyTorch library. SHM's running time could be further improved with proper parallelization.

## 4.4 MODEL ANALYSIS AND ABLATION STUDY

**Choice of Calibration** Prop. 5 suggests that selecting random $\theta_t \in \mathbb{R}^H$ in Eq. 13 will reduce the dependencies between $C_t$, bringing $\prod_{t=1}^T C_t$ closer to **1** to avoid gradient issues. In this section, we

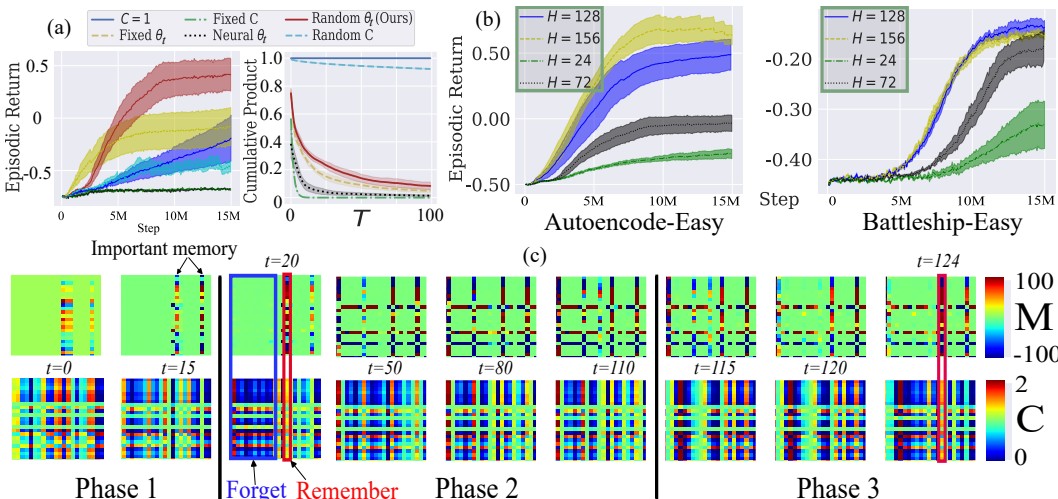

Figure 3: (a) Left: Return of calibration designs over 3 runs; Right: Calibration matrix cumulative product over 100 episodes. (b) Return of memory sizes $H$ on Autoencode-Easy (left) and Battleship-Easy (right). (c) Memory ($M$, top) and calibration ($C$, bottom) matrices over timesteps in Visual Match: SHM erases memory that are no longer required and strengthens the important ones.

empirically verify that by comparing our proposed Random $\theta_t$ with the following designs: $C = \mathbf{1}$, no calibration is used; *Random $C$*, where a random calibration matrix is sampled from normal distribution at each timestep, having $\mathbb{E}\left[\prod_{t=1}^{T} z_t^{m,k}\right] = 1$ under mild assumptions, but preventing learning meaningful calibration rules; *Fixed $C$*, a simpler method for learning calibration, but prone to gradient problems (Prop. 3); *Fixed $\theta_t$*, where we learn fixed parameter $\theta_t$, which is proven to be less effective than Random $\theta_t$ in reducing linear dependencies between timesteps (Prop. 5); Neural $\theta_t$, where $\theta_t = FFW(x_t)$, generated by a feedforward neural network like mLSTM, but with no guarantee of reducing timestep dependencies. We trained RL agents using the above designs of the calibration matrix with $H = 72$ on the Autoencode-Easy task and present the learning curves in Fig. 3 (a, left). The results show that our proposed Random $\theta_t$ outperforms the other baselines by a substantial margin, with at least a 30% improvement in performance. This confirms the effectiveness of our calibration design in enhancing the final results.

**Vanishing Behavior** In practice, exploding gradients can be mitigated by clipping. Thus, we focus on the vanishing gradient, which depends on the cumulative product $\mathcal{C}_j = \prod_{t=1}^{j} C_t$. Our theory suggests that Random $\theta_t$ should be less susceptible to the vanishing phenomenon compared to other methods such as Fixed $C$, Fixed $\theta_t$ and Neural $\theta_t$. To verify that, for each episode, we compute the average value of elements in the matrix $\mathcal{C}_j$ that are smaller than 1 ($\overline{\mathcal{C}_j}\,[< 1]$), as those larger than 1 are prone to exploding and are not appropriate for averaging with the vanishing values. We plot $\overline{\mathcal{C}_j}\,[< 1]$ for $j = 1, 2, ...100$ over 100 episodes in Fig 3 (a, right).

The results are consistent with theoretical predictions: Fixed $C$ leads to rapid vanishing of the cumulative product in just 10 steps. Neural $\theta_t$ also perform badly, likely due to more complex dependencies between timesteps because $z_t^{m,k}$ now becomes non-linear function of $x_t$, causing $\prod_{t=1}^{T} z_t^{m,k}$ to deviate further from 1. While Fixed $\theta_t$ is better than Neural $\theta_t$, it still exhibits quicker vanishing compared to our approach Random $\theta_t$. As expected, Random $C$ shows little vanishing, but like setting $C = 1$, it fails to leverage memory calibration, resulting in underperformance (Fig. 3 (a, left)). Random $\theta_t$, although its $\overline{\mathcal{C}_j}$ also deviates from 1, shows the smallest deviation among the calibration learning approaches. Additionally, the vanishing remain manageable after 100 timesteps, allowing gradients to still propagate effectively and enabling the calibration parameters to be learned.

**Memory Size** The primary hyperparameter of our method is $H$, determining the memory capacity. We test SHM with $H \in \{24, 72, 128, 156\}$ on the Autoencode-Easy and Battleship-Easy tasks. Fig. 3 (b) shows that larger memory generally results in better performance. In terms of speed,

the average batch inference times (in milliseconds) for different $H$ values are 1.7, 1.8, 1.9, and 2.1, respectively. We choose $H = 128$ for other POPGym tasks to balance performance and speed.

**Forgetting and Remembering** We study the learned calibration strategy of SHM on Visual Match with 15,100, and 10 steps in Phase 1, 2 and, 3, respectively. We sample several representative $M_t$ and $C_t$ from 3 phases and visualize them in Fig. 3 (c). In Phase 1, the agent identifies the color code and stores it in $M$, possibly in two columns of $M$, marked as "important memory". In Phase 2, unimportant memory elements are erased where $C_t \approx 0$ (e.g., those within the blue rectangle). However, important experiences related to the Phase 1's code are preserved across timesteps until Phase 3 (e.g., those within the red rectangle where $C_t \gtrsim 1$), which is desirable.

## 5 RELATED WORKS

Numerous efforts have been made to equip RL agents with memory mechanisms for handling POMDPs. Memory models can be broadly categorized into vector-based and matrix-based approaches. Vector-based memory, like RNNs (Elman, 1990), processes inputs sequentially and stores past inputs in their hidden states. While RNNs are slower to train, they are efficient during inference. Advanced variants, such as GRU and LSTM, have shown strong performance in POMDPs, often outperforming more complex RL methods (Ni et al., 2022; Morad et al., 2023). Others focus on the use of permutation-invariant sequence models in meta-reinforcement learning, demonstrating their advantage even without task inference objectives (Beck et al., 2024a). Recently, faster alternatives like convolutional and structured state space models (SSM) have gained attention (Bai et al., 2018; Gu et al., 2020), though their effectiveness in RL is still under exploration. Initial attempts with models like S4 underperformed in POMDP tasks (Morad et al., 2023), but improved SSM versions using S5, S6 or world models have shown promise (Lu et al., 2024; Gu & Dao, 2023; Samsami et al., 2024). Despite these advancements, vector-based memory is limited, as compressing history into a single vector makes it challenging to scale for high-dimensional memory space.

Matrix-based memory, on the other hand, offers higher capacity by storing history in a matrix but at the cost of increased complexity. Attention-based models, such as Transformers (Vaswani et al., 2017), have largely replaced RNNs in SL, also delivering good results in standard POMDPs (Parisotto et al., 2020). However, their quadratic memory requirements limit their use in environments with long episodes. Empirical studies have also shown that Transformers struggle with long-term memorization and credit assignment tasks (Ni et al., 2024). While classic memory-augmented neural networks (MANNs) demonstrated good performance in well-crafted long-term settings (Graves et al., 2016; Hung et al., 2019; Le et al., 2020), they are slow and do not scale well in larger benchmarks like POPGym (Morad et al., 2023). New variants of LSTM (Beck et al., 2024b), including those based on matrices, have not been tested in reinforcement learning settings and lack theoretical grounding to ensure stability.

Simplified matrix memory models (Katharopoulos et al., 2020; Schlag et al., 2021), offer scalable solutions but have underperformed compared to simple RNNs in the POPGym benchmark, highlighting the challenges of designing efficient matrix memory for POMDPs. Recently, Fast and Forgetful Memory (FFM, (Morad et al., 2024)), incorporating inductive biases from neuroscience, has demonstrated better average results than RNNs in the benchmark. However, in the most memory-intensive environments, the improvement remains limited. Compared to our approach, these matrix-based memory methods lack a flexible memory calibration mechanism and do not have robust safeguards to prevent numerical and learning issues in extremely long episodes.

## 6 DISCUSSION

In this paper, we introduced the Stable Hadamard Framework (SHF) and its effective instance, the Stable Hadamard Memory (SHM), a novel memory model designed to tackle the challenges of dynamic memory management in partially observable environments. By utilizing the Hadamard product for memory calibration and update, SHM provides an efficient and theoretically grounded mechanism for selectively erasing and reinforcing memories based on relevance. Our experiments on the POPGym and POMDP benchmarks demonstrate that SHM significantly outperforms state-of-the-art memory-based models, particularly in long-term memory tasks, while being fast to execute.

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

APPENDIX

## A    THEORETICAL RESULTS

### A.1    CLOSED-FORM MEMORY UPDATE

Given

$$M_t = M_{t-1} \odot C_t + U_t \tag{14}$$

then

$$M_t = M_0 \prod_{i=1}^{t} C_i + \sum_{i=1}^{t} U_i \odot \prod_{j=i+1}^{t} C_j \tag{15}$$

*Proof.* We prove by induction.

**Base case:** for $t = 1$, the equation becomes for both Eqs. 14 and 15:

$$M_1 = M_0 \cdot C_1 + U_1$$

Thus, the equation holds for $t = 1$.

**Inductive hypothesis:** Assume the equation holds for $t = n$:

$$M_n = M_0 \prod_{i=1}^{n} C_i + \sum_{i=1}^{n} U_i \odot \prod_{j=i+1}^{n} C_j$$

**Inductive step:** We now prove the equation holds for $t = n + 1$. From the update rule in Eq. 14:

$$M_{n+1} = M_n \odot C_{n+1} + U_{n+1}$$

Substitute $M_n$ using the inductive hypothesis:

$$M_{n+1} = \left( M_0 \prod_{i=1}^{n} C_i + \sum_{i=1}^{n} U_i \odot \prod_{j=i+1}^{n} C_j \right) \odot C_{n+1} + U_{n+1}$$

$$= M_0 \prod_{i=1}^{n+1} C_i + \sum_{i=1}^{n} U_i \odot \prod_{j=i+1}^{n+1} C_j + U_{n+1}$$

$$= M_0 \prod_{i=1}^{n+1} C_i + \sum_{i=1}^{n+1} U_i \odot \prod_{j=i+1}^{n+1} C_j$$

This matches the form of the closed-form Eq. 15 for $t = n + 1$, completing the proof by induction.
$\square$

### A.2    PROPOSITION 3

**Definition 6.  Critical Memory Gradients**: In the Hadamard Memory Framework, we define the critical memory gradients of memory rules as follows:

$$G_1(i, t, \theta) = \frac{\partial \prod_{j=i+1}^{t} C_\theta(x_j)}{\partial \theta} = \sum_{j=i+1}^{t} \frac{\partial C_\theta(x_j)}{\partial \theta} \odot \prod_{k=i+1, k \neq j}^{t} C_\theta(x_k)$$

$$G_2(i, t, \theta) = \prod_{j=i+1}^{t} C_\theta(x_j)$$

Now we can proceed to the proof for Proposition 3.

*Proof.* In the case $C_\theta(x_t) = \theta \in \mathbb{R}^{H \times H}$, the critical gradients read:

$$G_1(i, t, \theta) = \sum_{j=i+1}^{t} \prod_{k=i+1, k \neq j}^{t} \theta$$
$$= (t - i) \theta^{t-i-1}$$

$$G_2(i, t, \theta) = \theta^{t-i}$$

If $\exists \, 0 \leq m, k < H$ s.t. $|\theta[m, k]| > 1$, $G_{1,2}(i, t, \theta)[m, k] \sim o\left(\theta[m, k]^{t-i-1}\right)$, and thus $\to \infty$, i.e., numerical problem arises. Let $\|\cdot\|$ denote the infinity norm, we also have:

$$\|G_1(i, t, \theta)\| = \left\|(t - i) \theta^{t-i-1}\right\| \leq (t - i) \|\theta\|^{t-i-1}$$

$$\|G_2(i, t, \theta)\| = \left\|\theta^{t-i}\right\| \leq \|\theta\|^{t-i}$$

Note that if $\forall m, k \, |\theta[m, k]| < 1$, $\|\theta\| < 1$, both terms become 0 as $t - i$ increases, thus learning problem always arises. In conclusion, in this case, to avoid both numerical and learning problems, $\forall m, k \, |\theta[m, k]| = 1$, which is not optimal in general. $\qquad \square$

## A.3 PROPOSITION 4

*Proof.* Let $z_t^{m,k} = u_t^m v_t^k$ where $z_t^{m,k} = (\theta_t \otimes v_c(x_t))[m, k]$, $u_t^m = \theta_t[m]$ and $v_t^k = v_c(x_t)[k]$. Using assumption (i) and (ii), $v_t^k$ is a Gaussian variable, i.e., $v_t^k \sim \mathcal{N}(0, \mu_t^k)$. By definition, $u_t^m$ is a categorical random variable that can take values $\{\theta[m, l_t]\}_{l_t=1}^{L}$ with equal probability $1/L$. For now, we drop the subscripts $m$, $k$ and $t$ for notation ease. The PDF of $z = uv$ can be expressed as a mixture distribution since $u$ is categorical and can take discrete values $\{u_l\}_{l=1}^{L}$. The PDF of $z$, denoted as $f(z)$, is given by:

$$f(z) = \sum_{l=1}^{L} P(u = u_l) f_{u_l v}(z)$$
$$= \frac{1}{L} \sum_{l=1}^{L} f_{u_l v}(z)$$

where $f_{u_l v}(z)$ is the PDF of $u_l v$, and $u_l$ is a constant for each $l$. Thus, $f_{u_l v}(z)$ is the scaled PDF of $u$:

$$f_{u_l v}(z) = \frac{1}{|u_l|} f_v\left(\frac{z}{u_l}\right)$$

Since $v \sim \mathcal{N}(0, \mu)$, the PDF of $v$, denoted as $f_v(x)$, is symmetric about 0, we have:

$$f_{u_l v}(z) = \frac{1}{|u_l|} f_v\left(\frac{z}{u_l}\right) = \frac{1}{|u_l|} f_v\left(\frac{-z}{u_l}\right) = f_{u_l v}(-z).$$

This shows that $f_{u_l v}(z)$ is symmetric around 0 for each $l$. Therefore, the PDF $f(z)$ is also symmetric:

$$f(z) = \frac{1}{L} \sum_{l=1}^{L} f_{u_l v}(z) = \frac{1}{L} \sum_{l=1}^{L} f_{u_l v}(-z) = f(-z)$$

Since $\tanh$ is an odd function and the PDF of $z^{m,k}$ is symmetric about 0, $\mathbb{E}\left[\tanh\left(z^{m,k}\right)\right] = 0$ and thus $\mathbb{E}\left[1 + \tanh\left(z^{m,k}\right)\right] = 1$. Finally, using assumption (iii), $\mathbb{E}\left[\prod_{t=1}^{T} z_t^{m,k}\right] = \prod_{t=1}^{T} \mathbb{E}\left[z_t^{m,k}\right] = 1$. $\qquad \square$

| Task | Input type | Policy/Value networks | RL algorithm | Batch size |
|------|------------|----------------------|--------------|------------|
| Meta-RL | Vector | 3-layer FFW + 1 Memory layer $(128, 128, 128, H)$ | SAC | 32 |
| Credit Assignment | Image | 2-layer CNN + 2-layer FFW + 1 Memory layer $(128, 128, H)$ | SAC-D | 32 |
| POPGym | Vector | 3-layer FFW + 1 Memory layer $(128, 64, H, 64)$ | PPO | 65,536 |

Table 2: Network architecture shared across memory baselines.

### A.4  PROPOSITION 5

*Proof.* Without loss of generality, we can drop the indice $m$ and $k$ for notation ease. Since $\theta_t \sim \mathcal{U}(\theta)$, it is reasonable to assume that each of $\{u_t, u_{t'}\}, \{u_t, v_t\}, \{u_t, v_{t'}\}, \{u_{t'}, v_t\}, \{u_{t'}, v_{t'}\}$ are independent. In this case, let us denote $X = u_t v_t$ and $X' = u_{t'} v_{t'}$, we have

$$
\begin{aligned}
\mathrm{Cov}(X, X') &= \mathrm{Cov}(u_t v_t, u_{t'} v_{t'}) \\
&= \mathbb{E}[u_t v_t \cdot u_{t'} v_{t'}] - \mathbb{E}[u_t v_t] \cdot \mathbb{E}[u_{t'} v_{t'}] \\
&= \mathbb{E}(u_t v_t)\mathbb{E}(u_{t'} v_{t'}) - \mathbb{E}(u_t)\mathbb{E}(v_t)\mathbb{E}(u_{t'})\mathbb{E}(v_{t'}) \\
&= \mathbb{E}(u_t)\mathbb{E}(v_t)[\mathbb{E}(u_{t'} v_{t'}) - \mathbb{E}(u_{t'})\mathbb{E}(v_{t'})] \\
&= \mathbb{E}[u_t]\mathbb{E}[u_{t'}]\mathrm{Cov}(v_t, v_{t'})
\end{aligned}
$$

The variances read:

$$
\mathrm{Var}(X) = \mathrm{Var}(u_t v_t) = \mathbb{E}[u_t^2 v_t^2] - \mathbb{E}[u_t v_t]^2 = \mathbb{E}[u_t^2]\left(\mathbb{E}[v_t^2] - \mathbb{E}[v_t]^2\right) = \mathbb{E}[u_t^2] \cdot \mathrm{Var}(v_t)
$$

Similarly:

$$
\mathrm{Var}(X') = \mathrm{Var}(u_{t'} v_{t'}) = \mathbb{E}[u_{t'}^2] \cdot \mathrm{Var}(v_{t'})
$$

Given $\rho$ as the Pearson Correlation Coefficient, consider the ratio:

$$
\begin{aligned}
\frac{|\rho(X, X')|}{|\rho(v_t, v_{t'})|} &= \frac{\frac{|\mathbb{E}[u_t]\mathbb{E}[u_{t'}]\mathrm{Cov}(v_t, v_{t'})|}{\sqrt{\mathbb{E}[u_t^2]\cdot\mathrm{Var}(v_t)\cdot\mathbb{E}[u_{t'}^2]\cdot\mathrm{Var}(v_{t'})}}}{\frac{|\mathrm{Cov}(v_t, v_{t'})|}{\sqrt{\mathrm{Var}(v_t)\cdot\mathrm{Var}(v_{t'})}}} \\
&= \frac{|\mathbb{E}[u_t]\mathbb{E}[u_{t'}]|}{\sqrt{\mathbb{E}[u_t^2]\mathbb{E}[u_{t'}^2]}}
\end{aligned}
$$

By the independence of $u_t$ and $u_{t'}$ and the Cauchy-Schwarz inequality:

$$
|\mathbb{E}[u_t]\mathbb{E}[u_{t'}]| = |\mathbb{E}[u_t u_{t'}]| \le \sqrt{\mathbb{E}[u_t^2]\mathbb{E}[u_{t'}^2]},
$$

which implies

$$
\frac{|\rho(u_t v_t, u_{t'} v_{t'})|}{|\rho(v_t, v_{t'})|} \le 1 \iff |\rho(u_t v_t, u_{t'} v_{t'})| \le |\rho(v_t, v_{t'})|
$$

The equality holds when $u_t = \beta u_{t'}$. $\qquad\square$

### B  DETAILS ON METHODOLOGY

In this section, we describe RL frameworks used across experiments, which are adopted exactly from the provided benchmark. Table 2 summarizes the main configurations. Further details can be found in the benchmark papers (Ni et al., 2022; Morad et al., 2023).

**Algorithm 1:** Theoretical Parallel Hadamard Memory Framework.

**Input:** $M_0 \in \mathbb{R}^{B \times H \times H}, C \in \mathbb{R}^{B \times T \times H \times H}, U \in \mathbb{R}^{B \times T \times H \times H}$
**Operator:** $\oplus$ parallel prefix sum, $\otimes$ parallel prefix product, $\odot$ Hadamard product
**Ouput:** $M = \{M\}_{t=1}^{n} \in \mathbb{R}^{B \times T \times H \times H}$

```
     /* Parallel prefix product along T. Complexity:  O(log(t)).      */
1  Cp = ⊗(C, dim = 1)
     /* Concatenation along T. Complexity:  O(1).                     */
2  D = concat([M0, C], dim = 1)
     /* Parallel prefix product along T. Complexity:  O(log(t)).      */
3  Dp = ⊗(D, dim = 1)
     /* Parallel Hadamard product (Cp ≠ 0).  Complexity:  O(1).       */
4  E = U ⊙ 1/Cp
     /* Parallel prefix sum along T. Complexity:  O(log(t)).          */
5  Ep = ⊕(E, dim = 1)
     /* Parallel sum.  Complexity:  O(1).                             */
6  M = Dp[:, 1 :] + Ep
```

| Task | URL | License |
|------|-----|---------|
| Meta-RL | https://github.com/twni2016/pomdp-baselines | MIT |
| Credit Assignment | https://github.com/twni2016/Memory-RL | MIT |
| POP-Gym | https://github.com/proroklab/popgym | MIT |

Table 3: Benchmark repositories used in our paper.

## C  DETAILS OF EXPERIMENTS

We adopt public benchmark repositories to conduct our experiments. The detail is given in Table 3

### C.1  SAMPLE-EFFICIENCY IN META REINFORCEMENT LEARNING AND LONG-TERM CREDIT ASSIGNMENT DETAILS

We report the choice of memory hyperparameter $H$ in Table. 4. Due to the lack of efficient parallel processing in the codebase from Ni et al. (2022), running experiments with larger memory sizes is prohibitively slow. As a result, we were only able to test models with 512 memory elements, limiting the potential performance of our SHM. This constraint contrasts with the POPGym benchmark with better parallel processing, where our method scales comfortably with larger memory sizes, as demonstrated later in C.2.

We also report the running time and final performance for Visual Match and Key-to-Door experiments for different episode lengths. The results in Table 5 demonstrate that SHM can scale reasonably in both computation efficiency and performance as the task complexity increases.

### C.2  POPGYM HARDEST GAMES DETAILS

In this experiment, we tuned the memory model size to ensure optimal performance. Specifically, for GRU, we tested hidden sizes of 256, 512, 1024, and 2048 on the Autoencode-Easy task. Increasing GRU's hidden size did not lead to performance gains but resulted in a significant rise in computation

| Model | GRU | FWP | GPT-2* | S6 | mLSTM | FFM | SHM (Ours) |
|-------|-----|-----|--------|-----|-------|-----|------------|
| $H$ | 512 | 24 | 512 | 512 | 24 | 128 | 24 |
| Memory elements | 512 | 512 | $512 \times T$ | 512 | 512 | 512 | 512 |

Table 4: Memory dimension for Meta-RL and Credit Assignment tasks. GRU and S6 use vector memory of size $H$. GPT-2 does not have a fixed size memory and attend to all previous $T$ timesteps in the episode. $H = 512$ is the dimension of Transformer's hidden layer. FFM's memory shape is $2 \times m \times c$ where $c = 4$, $H = m = 128$. FWP and SHM's memory shape is $H \times H$.

| Episode Length | Total Training Time (hours) | | Success Rate (%) Visual Match | | Success Rate (%) Key To Door | |
|---|---|---|---|---|---|---|
| | SHM | GPT-2 | SHM | GPT-2 | SHM | GPT-2 |
| 100 | 20 | 18 | 100 | 83 | 100 | 59 |
| 250 | 27 | 23 | 100 | 72 | 78 | 21 |
| 500 | 41 | 36 | 100 | 47 | 55 | 22 |

Table 5: Average running time and success rate across episode lengths.

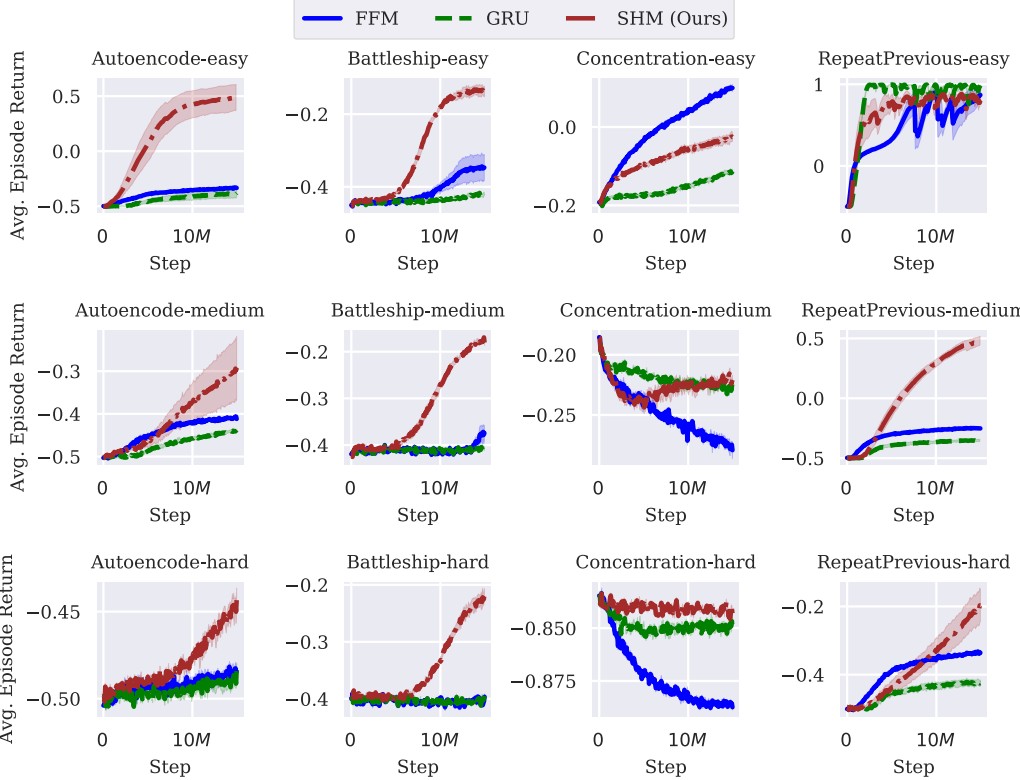

Figure 4: POPGym learning curves: Mean $\pm$ std. over 3 runs.

cost. For instance, at $H = 2048$, the average batch inference time was 30 milliseconds, compared to 1.6 milliseconds at $H = 256$. Thus, we set GRU's hidden size to 256, as recommended by the POPGym documentation. For FFM, we set the context size to $c = 4$, as the authors suggest that a larger context size degrades performance. We tuned the trace size $m \in \{32, 128, 256, 512\}$ on the Autoencode-Easy task and found that $m = 128$ provided slightly better results, so we used this value for all experiments. For our method, we tested $H \in \{24, 72, 128, 156\}$ on the same task and observed that larger values of $H$ led to better performance, as shown in the ablation study in Sec. 4.4. However, to balance runtime, memory cost, and performance, we set $H = 128$ for all POPGym experiments. The learning curves are given in Fig. 4. The final results are reported fully in Table 6.

## D MORE RELATED MEMORY MODELS

Recently, Beck et al. (2024b) have proposed matrix-based LSTM (mLSTM):

$$M_t = \mathtt{f}\left(x_t\right) M_{t-1} + \mathtt{i}\left(x_t\right) v\left(x_t\right) \otimes k\left(x_t\right) \tag{16}$$

where $\mathtt{f}$ and $\mathtt{i}$ are the forget and input gates, respectively.

| Task | Level | Linear Transformer[*] | GRU | FFM | SHM (Ours) |
|---|---|---|---|---|---|
| | Easy | -44.7±1.4 | -37.9±7.7 | -32.7±0.6 | **49.5±23.3** |
| Autoencode | Medium | -47.8±0.2 | -43.6±3.5 | -32.7±0.6 | **-28.8±14.4** |
| | Hard | -48.1±0.1 | -48.1±0.7 | -47.7±0.5 | **-43.9±0.9** |
| | Easy | -41.3±0.5 | -41.1±1.0 | -34.0±7.1 | **-12.3±2.4** |
| Battleship | Medium | -39.2±0.3 | -39.4±0.5 | -37.1±3.1 | **-16.8±0.6** |
| | Hard | -38.4±0.2 | -38.5±0.5 | -38.8±0.3 | **-21.2±2.3** |
| | Easy | -18.5±0.2 | -10.9±1.0 | **10.7±1.2** | -1.9±2.4 |
| Concentration | Medium | -18.6±0.2 | -21.4±0.5 | -24.7±0.1 | **-21.0±0.8** |
| | Hard | **-83.0±0.1** | -84.0±0.3 | -87.5±0.5 | -83.3±0.1 |
| | Easy | 6.0±4.0 | **99.9±0.0** | 98.4±0.3 | 88.9±11.1 |
| RepeatPrevious | Medium | -46.8±1.1 | -34.7±1.7 | -24.3±0.4 | **48.2±7.2** |
| | Hard | -48.5±0.3 | -41.7±1.8 | -33.9±1.0 | **-19.4±9.9** |
| Average | All | -39.1±0.4 | -28.4±1.3 | -24.2±1.2 | **-5.1±6.3** |

Table 6: PopGym: Mean return $\pm$ std. ($\times 100$) at the end of training over 3 runs. The range of return ($\times 100$) is $[-100, 100]$. $*$ is reported from Morad et al. (2023).

