# OpenReview forum: "Stable Hadamard Memory: Revitalizing Memory-Augmented Agents for Reinforcement Learning"
_ICLR.cc/2025/Conference — ICLR 2025 Poster_

### Official Review · Reviewer_3seu · 2024-10-27

**Soundness:** 4
**Presentation:** 3
**Contribution:** 3
**Rating:** 6
**Confidence:** 4

**Summary:**

This paper introduce"SHM"model, spcefically solving memory management problem in RL. SHM use a Hadamard product-based memory update mechanism to dynamically retain important information and discard irrelevant data, aiming to enhance memory stability and efficiency. The authors propose SHM as a solution to the limitations of existing memory models, such as numerical instability and inefficient memory retention, particularly in complex and long-horizon tasks. Experimental results demonstrate SHM’s superior performance compared to state-of-the-art memory models across multiple benchmarks, including meta-reinforcement learning and the POPGym suite, indicating its potential as a robust memory mechanism for RL applications.

**Strengths:**

This paper's got some fresh ideas that really matter in the real world. First off, they've come up with this nifty thing called the Stable Hadamard Memory (SHM) model. It's like a brain upgrade for AI in situations where it can't see everything. The unique twist is this memory update method that uses something called the Hadamard product. It's all about keeping the good memories and dumping the useless ones, which makes the AI's memory super stable and efficient.

But that's not all. The paper really shows off the SHM's chops by putting it through its paces in a bunch of tests.  It outperforms the current bigwigs in memory models by a mile. This model's got the chops to handle those tough, long-term memory tasks that other models can't touch, especially when it comes to memory-intensive tasks in reinforcement learning.

So, in a nutshell, the research in this paper could really move the needle in the field of reinforcement learning.

**Weaknesses:**

In reinforcement learning, especially when dealing with "partially observable" environments, effective memory management is really crucial. As task complexity increases, memory demand also skyrockets, and challenges grow alongside it. So, discussing how models "hold up" as tasks become more complex and expansive is pretty important.

**Questions:**

Hopefully, author interpret HaM in aspect of  "scaling law".

① Performance vs. Memory Size: How does the performance of the Hadamard Memory model scale with increases in memory size? Is there an optimal memory size at which the model's performance peaks?

② Impact of Task Complexity: As task complexity increases, how do the computational requirements and performance of the model change? Can the authors provide metrics to measure the impact of task complexity on model performance?

③ Scalability with Episode Length: Does the Hadamard Memory model maintain stable and efficient updates in longer episodes? Have the authors considered the model's stability and efficiency over extended periods?

---

> ### Author Response · Authors · 2024-11-20
> **Response to Reviewer 3seu**
>
> Thank you for your thoughtful review. We address your concerns point by point below.
>
> ### Weakness
> In the original manuscript, we have presented results and discussions regarding the change in performance across task complexity levels. The answers to your questions below will provide more details.
>
> ### Questions
> 1.  We tested SHM on PopGym tasks with different memory sizes $\( H \)$, and the results are shown in Fig. 3b. Overall, performance improves as  $H$  increases; however, in one task, performance peaks at $H = 128$. Thus, we selected $H = 128$ for other PopGym tasks to balance both performance and computational efficiency.
> 2.  In these POMDP tasks, the episode length often determines task complexity, with longer episodes typically being more challenging. We evaluated performance (measured by success rate) as episode length increased and reported the results in Fig. 2. Additionally, in the meta-RL setting, we studied task complexity by varying the train-test task ratio (lower ratios indicate higher complexity). The performance under different train-test ratios is presented in Fig. 1. Overall, as task complexity increases, the performance of all methods declines. However, our SHM maintains reasonable performance and clearly outperforms other approaches. Regarding computational requirements, we have added Table 5 in the appendix of the revised manuscript to show the running time of our method vs GPT-2 as task complexity changes. As shown, the running time of our method remains comparable to that of the other baseline (see the table in our answer to your question 3). Please note our current implementation of SHM is naive without any parallel support while GPT-2 implementation has Pytorch-optimized attention.
>
> 3.  Yes, in Fig. 2, we reported the performance of SHM over extended episodes, including lengths of 250 and 500 steps. (The benchmark in [1] also only tested up to 500 steps for Key-to-Door tasks). Beyond this limit, testing requires significant computational and memory resources for all methods, which are beyond our current capacity. That said, we agree that it is valuable to present results with more diverse episode lengths. To address this, we have added results with an episode length of 100 in the appendix. These results further demonstrate the dominance of SHM over GPT-2 (best performer reported in [1]). Please see example comparison below for both time and performance as episode length increases for Key To Door (see more in Appendix Table 5).
> | Episode Length | SHM Total Training Time (hours) | GPT-2 Total Training Time (hours) | SHM Success Rate (%) | GPT Success Rate (%) |
> |----------------|---------------------------------|-----------------------------------|----------------------|-----------------------|
> | 100            | 20                              | 18                                | 100                  | 59                    |
> | 250            | 27                              | 23                                | 78                  | 21                    |
> | 500            | 41                              | 36                                | 55                  | 25                    |
>
>
>
>
> We hope these revisions adequately address your concerns. Please feel free to let us know if there is anything else we can clarify or enhance to improve your score on our paper.
>
> ### Reference
> [1] Tianwei Ni, Michel Ma, Benjamin Eysenbach, and Pierre-Luc Bacon. When do transformers shine in rl? decoupling memory from credit assignment. Advances in Neural Information Processing Systems, 36, 2024

---

> > ### Comment · Reviewer_3seu · 2024-11-24
> >
> > Thank you for addressing my questions and providing detailed responses. The explanations and additional insights have clarified the points I raised, and I am satisfied with the way the authors have addressed my concerns. I have no further issues with the submission.

---

### Official Review · Reviewer_SMQZ · 2024-10-30

**Soundness:** 3
**Presentation:** 2
**Contribution:** 3
**Rating:** 8
**Confidence:** 4

**Summary:**

The authors propose a relatively simple recurrent model for reinforcement learning under partial observability. Their model is parallelizable in time due to a linear recurrent state update. They demonstrate that their method outperforms prior work across various benchmarks, both with PPO SAC algorithms.

All in all, this is a good paper tackling an important problem with a novel approach and thorough experimentation. In my opinion, the biggest weakness is that the authors have not provided their code.

**Strengths:**

- The paper is nicely structured
- The authors provide a thorough literature review
- The authors' method is very straightforward to implement
- The derivation of the linear matrix C is much shorter and easier to read than the derivation of the A and B matrices from the State Space Models paper (and yet, the authors' method performs better)
- Rather than designing linear operations to keep inputs around in the recurrent state as long as possible (State Space Models), the authors take relatively novel approach to design a linear operation that has well-behaved gradients
- Randomly sampling $\theta$ to break the dependence between inputs is a novel and interesting idea
- The authors run numerous experiments, across multiple benchmarks and RL algorithms

**Weaknesses:**

- The authors do not provide code
- The writing is a bit awkward in places, hampering my understanding -- I think the paper would benefit from having another proofread
    - The initialization of $C$ is quite important (lines 259-264), but it took me a few reads to understand the process -- perhaps the authors can consider writing something like $\theta = [\theta_1, \theta_2, \dots, \theta_{L}]; \quad \theta_t \sim \mathcal{U}(\theta)$
    - Proposition 3 is poorly worded, and should be rewritten
    - Section 3.3 is misspelled (STABLE HADAMRAD MEMORY (SHM))

**Questions:**

- Why is $1 + \tanh(x)$ used instead of $\textrm{sigmoid}(x)$?
- How come the recurrent state is real-valued compared to methods like FFM? Did the authors try using complex-valued states to track phase information across sequences?
- Line 233: What is $m.k$?

---

> ### Author Response · Authors · 2024-11-20
> **Response to Reviewer SMQZ**
>
> Thank you for your positive review. We address your concerns point by point below.
>
> ### Weakness
> **"The authors do not provide code"**: Due to time constraints, we did not submit the code at the submission time. However, as mentioned in the original manuscript, we committed to release the code. In this revision, we have attached the code for SHM and the baselines in Popgym tasks. We plan to complete and release the full code upon publication.
>
> **"The initialization of ..."**: Thank you for your suggestions. We have revised this part accordingly.
>
> **"Proposition 3 is poorly worded ..."**: Thank you for pointing it out. We have changed it to:
>
> "Proposition 3. If calibration is enabled such that $C_{\theta}(x_{t})\neq\boldsymbol{1}$, and the calibration matrix remains fixed and independent of the input $x_{t}$ (i.e., $\forall t:C_{\theta}(x_{t})=\theta\in\mathbb{R}^{H\times H}$ ), this will lead to either numerical instability or learning difficulties"
>
> **"Section 3.3 is misspelled ..."**: Thank you, we have fixed it.
>
> ### Questions
>
> **"Why is 1+tanh⁡(x) ...."**: Because the proof in Appendix A.3 requires an odd function like tanh (see Line 753 in the original manuscript)
>
> **"How come the recurrent state ..."**: Thank you for your insightful comments. Our method does not utilize complex-valued states; instead, it employs real-valued recurrent states like standard models such as GRU. Incorporating complex-valued states to track phase information could be an orthogonal contribution to our approach. We believe that integrating this aspect could potentially enhance performance further. However, we will leave this exploration for future work.
>
> **"Line 233: ..."**: This is a typo. We have revised it to m,k. Here, m and k are the row and column indices of a matrix.

---

> ### Comment · Reviewer_SMQZ · 2024-11-20
>
> Thank you for the response. The authors have addressed all of my concerns, and I maintain that this paper should be accepted. I agree with the other reviewer, in that perhaps there is a more descriptive name than "Stable Hadamard Memory". But ultimately, this is up to the authors.

---

### Official Review · Reviewer_Av1T · 2024-11-03

**Soundness:** 3
**Presentation:** 2
**Contribution:** 3
**Rating:** 6
**Confidence:** 2

**Summary:**

This work addresses memory in partially observable RL environments. The authors state that current methods fail to capture relevant past information, making it difficult to retain important memories and know when to dynamically update them in response to changes in the environment. This issue is particularly significant for continual learning.

The authors introduce a Hadamard memory framework, suggesting that it performs memory calibration and linearly adjusts memory elements by maintaining a calibration matrix used to update these elements via an update matrix. The Hadamard product allows these elements to be learned using neural networks, and this method is well-supported in the literature. The memory matrix is then formulated using the standard policy gradient method, and the authors also demonstrate stability guarantees.

This approach is based on the Hadamard product applied to the memory matrix, enabling adaptive memory rules to be learned.

The method is evaluated on various Meta-RL tasks and challenging credit assignment tasks, with results showing dramatic improvements. The authors propose that this dynamic memory writing method facilitates more efficient credit assignment. They also evaluate the method on a set of partially observable gym environments, visually demonstrating the memory and calibration matrices.

**Strengths:**

* As said in my summary, the results show improvements on a broad set of tasks and appear to be extremely significant.

* The method is mathematically well motivated, justified well and theoretically guarantees are given. I especially appreciate the stability guarantees.  It appears to be sound to use this for updating agent memories.

* The credit assignment benefits are justified well and empirically shown.

* The work is very well motivated and many of the theoretical guarantees are strong and supported well.

Overall this work is solid and presents and interesting way of updating memory.  I currently support acceptance of this work.  I do not have high confidence in my review as this work is a bit orthogonal to the previous works I have done on partially observably environments.

**Weaknesses:**

* The paper is overly complex at times going into too many mathematical details.  I would like a very clear description of the benefits of using this update rule and why it empirically works well.

**Questions:**

Please see weaknesses.

---

> ### Author Response · Authors · 2024-11-20
> **Response to Reviewer Av1T**
>
> Thank you for your encouraging reviews. We address your concern below.
>
> **"The paper is overly complex ..."**: We have simplified the notation on $\theta$ and added a simpler and more detailed description at the end of Sec. 3 to highlight and explain the benefit of the memory update rule as follows:
>
> "The proposed update rule improves memory management and stability through several key benefits. The dynamic calibration matrix $\(C_t\)$ adjusts memory elements based on the input context, allowing the model to prioritize relevant data while forgetting outdated information. This ensures that transient details, like detour events, are discarded while critical information, like landmarks, is retained. Moreover, the random parameter selection in the calibration matrix breaks timestep dependencies and controls the expected gradient norm, preventing numerical instabilities like gradient explosion or vanishing. This stable update mechanism ensures consistent learning and robust performance over long episodes."
>
> We hope these changes have effectively addressed your concerns. Please let us know if there is anything further we can clarify or improve to support a higher evaluation.

---

### Official Review · Reviewer_1jQd · 2024-11-03

**Soundness:** 2
**Presentation:** 3
**Contribution:** 2
**Rating:** 6
**Confidence:** 3

**Summary:**

This work presents a memory-based architecture for partially observable RL, named Stable Hadamard Memory (SHM). SHM is evaluated on a range of POMDP environments and POPGym against recurrent and attention-based baselines, as well as some ablations and analysis.

**Strengths:**

The paper is well presented and the method is explained in detail. It is evaluated against a range of baselines and on multiple benchmarks.

**Weaknesses:**

1. The novelty of the method is not directly apparent from reading the paper. Stable Hadamard Memory is a confusing description for the method, as many existing RNNs, such as GRUs, make use of a Hadamard product as a core component. Furthermore, the Hadamard product is not the defining feature of the method, any more than addition. Instead, it would seem that the design of the calibration matrix, with random selection/dropout of update parameters, is the primary contribution of the approach. Based on this, more focus on this component in the paper and method naming would benefit the reader.

2. The method's update rule is highly similar to linear attention, which is cited but not discussed in the work. A discussion of how this method relates to linear attention and empirical evaluation against it would be compelling.

3. Due to the random selection of update parameters in SHM, the memory usage of SHM should be higher than methods without parameter dropout. Discussing this and adding experiments comparing SHM to baselines with the same memory usage would be useful.

4. The experiments have very few repeats (3 seeds) and it is unclear how some baselines were tuned.

5. A paper from RLC 2024 exploring memory aggregation in Meta-RL is not cited [1].

[1] Jacob Beck, Matthew Jackson, Risto Vuorio, Zheng Xiong, and Shimon Whiteson. Splagger: Split aggregation for meta-reinforcement learning. Reinforcement Learning Conference, 2024.

**Questions:**

* Typo in the title of Section 3.3 - Stable Hadamrad Memory.

* How were the baselines tuned for each of the experiments? I could only find partial detail in the appendix. What efforts were made to normalize computational and memory cost between baselines and SHM?

---

> ### Author Response · Authors · 2024-11-20
> **Response to Reviewer 1jQd (1/2)**
>
> Thank you for your constructive comments. We address your concerns point by point below.
>
> ### Weaknesses
> 1. Thank you for your insightful comments on the novelty and naming of our approach. “Hadamard Memory” is a unified framework for matrix-based memory, distinct from vector-based memories in RNNs and GRUs. Although "Matrix Hadamard Memory" might be clearer, we chose a more concise name.
> While the Hadamard product is used in other architectures, we leverage it for memory calibration, enabling matrix-level memory updates. Please note that a key contribution of our work is identifying this unified framework, and analyzing its theoretical properties, and limitations (Sections 3.1-3.2). Based on the analysis, we propose a solution to improve the Hadamard Memory Framework. We named it “Stable Hadamard Memory” to emphasize our focus on stability, particularly through the calibration matrix’s random parameter selection. We used Section 3.3 to write about this stability component. To make it clearer,  we have revised the introduction and add more descriptions to Sec.3 ending to clarify the contributions regarding designing the calibration matrix with random parameter selection for stable updates.
>
> 2.  Although our approach might appear similar to linear attention in Linear Transformer [1], there is a clear distinction between our Hadamard Memory Framework (Eq. 7-8) and linear attention (Eq. 5):
> - Linear attention lacks the calibration matrix C, which is central to our framework, as explained in Section 3.1.
> - Linear attention does not include the update gate η​, which enhances memory control in our method.
>
> We have clarified these points in Sec. 3.1 of the revision. We did not initially report the performance of Linear Transformer because, according to the Popgym benchmark [2], it underperforms against baselines like GRU and FFM. As a result, we prioritized comparisons with stronger or more recent baselines (for your reference, we have added the reported results of Linear Transformer to Appendix Table 6 in this revision).
> Moreover,  we have also added mLSTM [3], a recent and more related matrix-based LSTM variant, to our experiments. New results in Fig. 1 and 2 demonstrate that SHM still significantly outperforms mLSTM. We also summarize the results between our SHM and mLSTM in the table below.
>
> | Task                     | SHM Success Rate (Mean ± Std) (%)    | mLSTM Success Rate (Mean ± Std) (%)   |
> |--------------------------|---------------------------------------|---------------------------------------|
> | Wind-50-150             | **82.45 ± 0.57**                    | 34.74 ± 48.01                        |
> | Wind-10-190             | **51.76 ± 5.38**                    | 24.89 ± 34.73                        |
> | Point Robot 50-150      | **93.33 ± 1.63**                    | 50.00 ± 17.32                        |
> | Point Robot 10-190      | **41.84 ± 11.84**                   | 8.05 ± 0.16                          |
> | Visual Match-250        | **100.00 ± 0.00**                   | 62.50 ± 7.50                         |
> | Visual Math-500         | **100.00 ± 0.00**                   | 65.00 ± 5.00                         |
> | Key-to-Door-250         | **77.50 ± 12.50**                   | 47.50 ± 22.50                        |
> | Key-to-Door-500         | **55.00 ± 0.00**                    | 45.00 ± 0.00                         |
>
>
> 3.  The random selection mechanism does not substantially increase memory requirements for storing the model's parameter. The additional parameters introduced by our calibration matrix $\( \theta \in \mathbb{R}^{L \times H} \)$, with $L = 128$ and $H$ typically set to 24, resulting in around 3,000 extra parameters—less than 3\% of the total memory used by the standard policy and  value networks in our experiments (see Appendix Table 2). Thus, the effect on memory usage is minimal, and comparable to other baselines in term of both number of parameters and model's memory capacity (see results in Sec. 4.1)
>
> 4. We followed the benchmark from Morad et al., (2023) [2] and trained each model 3 times per task. Please note that the RL tasks we used are long-term by design, with each run taking several days to complete. Given the large number of tasks (24 in total) and resource limitations, we believe that 3 runs are reasonable, and the results still show clear statistical differences between the methods. For details on the tuning process, we provided an explanation in Appendix C1 and C2 in the original manuscript. Basically, we focus on tuning $H$, the memory dimension.
>
> 5. Thank you for pointing it out. Although this paper does not focus on the matrix memory update aspect, it is indeed related.   We have included this paper in the related work of the revision.

---

> > ### Author Response · Authors · 2024-11-20
> > **Response to Reviewer 1jQd (2/2)**
> >
> > ### Questions
> >
> > **"Typo in the title ...":** Thank you for pointing out the typo. We have fixed it in the revision.
> >
> > **"How were the baselines tuned ..."**: Thank you for your questions. For all of our experiments, the baselines share the same RL framework provided by the benchmark. The only difference between the models is the memory layer, which we replace with various memory models. The key hyperparameter we tune for these memory models is $H$, which determines the number of memory elements. As explained in Appendix C.1, we normalized the number of memory elements across all baselines to 512 for the models in the POMDP benchmark [4]. We also experimented with different values of $H$ for the baselines examined in the PopGym benchmark, as reported in Appendix C2. For these RL tasks, as mentioned in Appendix C.2, increasing the number of parameters does not mean better performance.
> >
> > Regarding computational cost, the running time comparison is provided in Section 4.3 (last paragraph). Overall, we show that in both settings (where $H$ is fixed or tuned), SHM outperforms other methods. Although the non-parallel implementation of SHM is slower than GRU and FFM, the difference in speed is small, especially compared to other matrix memory like FFM. Finally, we have described memory cost in our answer to your question 3 above.
> >
> > We hope our responses have addressed your concerns. If you find our answers satisfactory, we would appreciate your consideration in increasing your score.
> >
> > ### References:
> >
> > [1] Katharopoulos, Angelos, Apoorv Vyas, Nikolaos Pappas, and François Fleuret. "Transformers are rnns: Fast autoregressive transformers with linear attention." In International conference on machine learning, pp. 5156-5165. PMLR, 2020.
> >
> > [2] Morad, Steven, Ryan Kortvelesy, Matteo Bettini, Stephan Liwicki, and Amanda Prorok. "POPGym: Benchmarking Partially Observable Reinforcement Learning." In The Eleventh International Conference on Learning Representations, 2023.
> >
> > [3] Beck, Maximilian, Korbinian Pöppel, Markus Spanring, Andreas Auer, Oleksandra Prudnikova, Michael Kopp, Günter Klambauer, Johannes Brandstetter, and Sepp Hochreiter. "xLSTM: Extended Long Short-Term Memory." arXiv preprint arXiv:2405.04517 (2024).
> >
> > [4] Tianwei Ni, Benjamin Eysenbach, and Ruslan Salakhutdinov. Recurrent model-free rl can be a
> > strong baseline for many pomdps. In International Conference on Machine Learning, pages
> > 16691–16723. PMLR, 2022.

---

> > > ### Comment · Reviewer_1jQd · 2024-11-24
> > >
> > > Thank you for the thorough response! I appreciate your adding a more explicit summary of the work's contributions, which were originally hard to gather from the paper, and the mLSTM results. I concur with other reviewers that the method is interesting and contains novel components and I am satisfied by the evaluation, so I have increased my score.

---

### Author Response · Authors · 2024-11-20
**General Response**

We sincerely appreciate the Reviewers' constructive and valuable feedback on our paper. We are pleased that they find our work “solid and well-motivated” (Reviewer Av1T), “novel and interesting” (Reviewer SMQZ), with “fresh ideas that really matter in the real world” (Reviewer 3seu). The Reviewers also appreciated our writing as “well presented” (Reviewer 1jQd) and “nicely structured” with “a thorough literature review” (Reviewer SMQZ). Finally, we are encouraged by the positive recognition of our experimental rigor, with all Reviewers noting the significant improvements across tasks and our thorough evaluation against a range of baselines and benchmarks.

We have addressed the remaining concerns in the individual reviews and made the following updates to the manuscript:
- Add baselines to the experiment (Fig. 1, 2 and Appendix Table 6)
- Add results on performance and running time as episode length varies (Appendix Table 5)
- Added more details to the introduction and Section 3 to clarify the benefits and contributions of our calibration matrix design using random parameter selection
- Fix writing errors, and typos, and simplify the math notations.

---

### Meta-Review · Area_Chair_s1YT · 2024-12-21

**Metareview:**

This paper proposes a memory-based architecture for solving POMDPs, centered on a "memory matrix" that can be updated via a Hadamard product, which addresses numerical stability issues and efficiency issues with prior architectures. Empirical results on meta-RL tasks and credit-assignment tasks show significant improvements.

The paper is simple and well motivated, and provides strong theoretical guarantees coupled with strong empirical performance. The reviewers appreciated the paper writing, finding it "nicely structured" and noting that the derivation was "easier to read" than the SSM paper.

The main reviewer suggestions were to discuss additional prior work (other methods that use a Hadamard product; linear attention; Splagger). In terms of experiments, reviewers encouraged the authors to release code and include additional random seeds.

Taken together, this paper contributes strong theoretical and empirical results which the reviewers unanimously voted to accept. I therefore recommend that the paper be accepted.

**Additional Comments On Reviewer Discussion:**

During the rebuttal, the authors added additional baseliens and ablations and revised parts of the paper to address reviewer concerns. The reviewers discussed the relationship with prior methods (e.g., linear transformer), the memory requirements of the proposed method, and details for hyperparameter tuning. The authors released code.

---

### Decision · Program_Chairs · 2025-01-22

Accept (Poster)